# Potential Role of Host Microbiome in Areca Nut-Associated Carcinogenesis and Addiction

**DOI:** 10.3390/molecules27238171

**Published:** 2022-11-24

**Authors:** Lihui Chen, Fulai Yuan, Sifang Chen, Xiong Li, Lingyu Kong, Wei Zhang

**Affiliations:** 1Department of Clinical Pharmacology, Xiangya Hospital, Central South University, 87 Xiangya Road, Changsha 410078, China; 2Institute of Clinical Pharmacology, Hunan Key Laboratory of Pharmacogenetics, Central South University, 110 Xiangya Road, Changsha 410078, China; 3Engineering Research Center of Applied Technology of Pharmacogenomics, Ministry of Education, 110 Xiangya Road, Changsha 410078, China; 4National Clinical Research Center for Geriatric Disorders, 87 Xiangya Road, Changsha 410008, China; 5Health Management Center, Xiangya Hospital, Central South University, Changsha 410078, China; 6Department of Neurosurgery, The First Affiliated Hospital of Xiamen University, Xiamen 361000, China; 7The First Affiliated Hospital of Guangdong Pharmaceutical University, Guangzhou 510060, China; 8Department of Radiology, Xiangya Hospital, Central South University, Changsha 410008, China

**Keywords:** areca nut, polyphenol, arecoline, gut microbiota, addiction, carcinogenicity

## Abstract

Areca nut (AN) is widely consumed all over the world, bringing great harm to human health and economy. Individuals with AN chewing are at high risk of cardiovascular disease and impaired immune system and metabolic system. Despite a growing number of studies having reported on the adverse effects brought by AN chewing, the exact mechanism of it is limited and the need for additional exploration remains. In recent years, the interaction between microorganisms, especially intestinal microorganism and host, has been extensively studied. AN chewing might disrupt the oral and intestinal microbiota communities through direct connect with the microbes it contains, altering PH, oxygen of oral and intestinal microenvironment, and disturbing the immune homeostasis. These mechanisms provide insights into the interplay between areca nut and host microbiota. Emerging studies have proposed that bidirectional interaction between polyphenols and intestinal microbes might play a potential role in the divergence of polyphenol, extracted from AN, among individuals with or without AN-induced cancer development and progression. Although some AN chewers have been aware of the harmful effects brought by AN, they cannot abolish this habit because of the addiction of AN. Increasing studies have tried to revealed that gut microbiota might influence the onset/development of addictive behaviors. Altogether, this review summarizes the possible reasons for the disturbance of host microbiota caused by areca nut chewing and clarifies the complex interaction between human microbiome and major constituents and the addiction and carcinogenicity of AN, tempting to provide novel insights into the development and utilization of it, and to control the adverse consequences caused by AN chewing.

## 1. Introduction

Approximately 10–20% of the world’s population use areca nut (AN) products in some form, with the most prevalent use in many regions in south Asia, south-east Asia, and the Asia Pacific region [1,2]. AN is also widely consumed, with or without tobacco, among the Australian, Canadian, European, and USA Indo-Asian immigrants [3]. AN has many clinical effects, such as dispelling nausea, improving cognitive performance, aiding digestion, inhibiting inflammation, fighting against parasitic infections and hypertension, and acting as an antidepressant as well as a substitute for cigarettes [4,5]. However, AN is the fourth most addictive substance used in the world, only surpassed by nicotine, alcohol, and caffeine [6,7]. AN has been identified as carcinogenic to humans (Group 1) by the International Agency for Cancer Research (IARC) [8]. There is much evidence that AN chewing is associated with oral, pharyngeal, and esophageal cancers [9,10,11]. It is also reported that AN chewing habit has detrimental effects on the metabolic system [12], increasing the risk of obesity [13], hyperglycemia [14], and, causing hypothyroidism [15] and vitamin D deficiency [16]. Chronic AN consumption also interferes with the immune system [17,18,19], causing suppression of T-cell activity and decreased release of cytokines [20]. AN usage also increases cardiovascular disease rate, including heart attack, coronary artery diseases [21,22] and paroxysmal supraventricular tachycardia (PSVT) [23]. Despite a growing number of studies that have reported the adverse effects brought by areca chewing, the exact mechanism remains to be explored. There are still about 600 million people, including children using betel around the world [6], causing great damage to human health and property.

As the second genome of the human body, the human microbiome has become an area of utmost interest, which is not a passive victim in many pathological processes, but a driver or stimulator in pathophysiological processes [24]. Microbiome refers to the community of microbes that reside in a defined environment, including bacteria, viruses, fungi, protozoa, along with their genes and genomes. The gastrointestinal tract is the most popular region for microbiota colonizing, followed by oral cavity [25]. Microbes residing in humans evolve with hosts and are susceptible to living habits, such as diet, tobacco, alcohol, and areca nut, which are causal factors of many disorders. Early studies implicated alterations in oral microbial composition in areca nut chewers with distinct oral premalignant lesions such as leukoplakia, erythroplakia, and submucous fibrosis [26,27], which contributes to oral cancers. Mei, et al. reported that AN seed polyphenol altered the composition of the gut microbiome [28]. In this review, we summarize the current understanding of the interaction of major constituents in areca nut and host microbiome and its involvement in the addiction and carcinogenesis of AN in an attempt to raise profound research questions that remain to be explored.

## 2. Microorganisms Contained in Areca Nut Alter Oral and Intestinal Microenvironment

AN products are easily available and the quality is not controlled very strictly because of their low prices [29]. AN is used at distinct stages of maturity in natural state or after processing in many forms, bringing different microbes to host microbiome. A pilot study, evaluating 12 samples of areca nut-containing chewing substances, found that wet gutka preparations were contaminated by *Escherichia coli* and *Enterobacteriacaea*. High levels of fungal aflatoxin (range: 0.43–1.84 mg/kg), a proven carcinogen, were identified in all samples [30]. Massive studies demonstrated the adverse health impacts of areca nut on systemic pathophysiological changes that might lead to disease, and that they are associated with the chemistry; metabolism; and pharmacology of polyphenols, tannins, trace elements, and areca alkaloids, specifically presenting in AN [31]. However, few studies investigated the microbes in areca nut in recent years, and this may be incriminated as causative factors in AN-chewing-associated diseases. More studies are required to be conducted to explore the microbes that distinct AN products contain. AN chewing is known to impair the host immune system, presenting suppression of T-cell activity and decreased release of cytokines [32]. Disruption of the immune balance might induce changes in the composition and function of the host microorganism. Furthermore, it is reasonable that chronic exposure to areca nut chewing is likely to favor specific bacterial colonization via altering oxygen, PH, and acid production of oral cavity. Hernandez, et al. demonstrated that current chewers had significantly elevated levels of *Streptococcus infantis* and various levels of distinct taxa of the *Actinomyces* and *Streptococcus* genera [26]. Zhou, et al. explored the effects of the Fuzhuan brick tea supplemented with different concentrations of ANs on gut microbiota in mice, and found that influence on intestinal microbial structure increased as the concentration of AN increased [33]. Altogether, mechanisms of areca nut chewing to influence host microbiota are through direct connect with the microbes it contains, altering the PH and oxygen of the oral and intestinal microenvironment, or through disturbing the immune homeostasis, and these mechanisms provide insights into the interplay between areca nut and host microbiota (Figure 1).

## 3. Major Constituents in Areca Nut

Areca nut, as a natural product, is composed of a variety of ingredients. Major constituents in AN include trace elements, polyphenols (flavanols and tannins), carbohydrates, alkaloids, proteins, fats, and crude fiber. Among them, the carcinogenic potential of AN is attributed to polyphenols and areca alkaloids [34,35]. The latter also contributes to addiction of AN [35]. Next, we reviewed the main components of areca nut that drives addiction and oncogenicity and highlighted the mechanisms by which microbiota and/or their microbial metabolites exert their action on the polyphenols and areca alkaloids of areca nut as promoters of the addictive and carcinogenic effects.

## 4. Carcinogens in Areca Nut 

A large body of literature found that habitual AN chewing was tightly associated with the occurrence and development of oral, esophageal, and pharyngeal cancers [9,10,11,36,37]. In addition, many studies revealed that long-term AN usage increased the risk of hepatocellular carcinoma (HCC) [38]. Chao, et al. found that AN chewing was a significant factor for non-muscle-invasive bladder cancer recurrence [39]. In-vitro arrays showed that areca nut extract treatment enhanced migration and invasion of head and neck squamous cell carcinoma (HNSCC) cells by upregulating cyclooxygenase-2 (COX-2)/vimentin expression, which is associated with poor survival of HNSCC patients [40]. Studies have reviewed the potential carcinogenic mechanism of alkaloids, while few studies have been conducted to explore the oncogenicity of areca nut polyphenols. 

### 4.1. Contradictory Role of Polyphenols in Cancers

Evidence is emerging on the toxicity of the dietary polyphenols extracted by areca nut [41]. Major polyphenols found in AN are tannins, catechins, flavonoids, safrole, and eugenol, among which, tannins, safrole, eugenol, and catechins have been proven to be carcinogens. Many experimental and preclinical studies proposed that polyphenol and tannin fractions of AN had a relevant role in BQ-induced cancer development and progression, mainly attributed to their immunomodulatory properties [42,43]. For example, reactive oxygen species (ROS) produced during the autoxidation of BN polyphenols in the saliva of chewing BQ is crucial in the initiation and promotion of oral cancer [44]. Also, incidences of esophageal cancer have been reported to be associated with consumption of tannins-rich foods such as BN, and carcinogenic activity of tannins might be related to components associated with tannins rather than tannins themselves, suggesting a potential role of gut microbiota in tannins carcinogenicity. Epidemiological studies found that a polyphenol-rich diet protected against cancer, as well as many short-term assays revealed that AN polyphenols and tannins were not mutagenic and, in fact, even had antimutagenic effects. Some literature reported that polyphenols favored the generation of ROS [45]. Contrasting this, AN polyphenols were reported to be able to form conjugates with carcinogens, to trap nitrite and ROS [46,47]. The prominent polyphenols in AN and their contradictory activities are summarized in Table 1. And reported major classes of polyphenols extracted from other plants are also showed in Table 2.

The underlying mechanism of these observed dual, and apparently contradictory, functions of AN polyphenols and tannins in the process of BQ-induced carcinogenesis remains to be explored.

### 4.2. Bidirectional Interaction between Polyphenols and Intestinal Microbes 

Polyphenols have an extremely low oral bioavailability and are almost unchanged when reaching the colon, and most of them are intensively catabolized by gut microbiota to a wide variety of new chemical structures that are often more active and better absorbed than the original phenolic compound passing hardly into the systemic blood circulation [59,60,61], and in turn, polyphenols could modulate oral and gut microbiota composition in host homeostasis and diseases. Several human and animal studies reported that polyphenols can elevate butyrate-producing bacterial and probiotics such as Lactobacillus and Bifidobacterium, while inhibiting opportunistic pathogenic or proinflammatory microbes. For example, bound polyphenol from foxtail millet bran can inhibit colitis-associated carcinogenesis by remodeling gut microbiota in a mice model [62]. The two-way interplay between microbial communities and polyphenols in the intestine is important for the latter to exert anticancer effects and might be the underlying mechanism for their contradictory and dual effects. Emerging studies have reported that diet polyphenols exert anti-obesity [63,64,65,66], anti-inflammatory, and anti-oxidant effects via modulating gut microbiota. For instance, Ho, et al. found that Heterogeneity in gut microbiota drive polyphenol metabolism that influences α-synuclein misfolding and toxicity [67]. Mei, et al. revealed that areca nut (areca catechu L.) seed polyphenol could ameliorate osteoporosis via altering gut microbiota to increase lysozyme expression and controlled the inflammatory reaction in estrogen-deficient rats [28]. Studies have shown that AN could supplement polyphenols such as chlorogenic acid, (+)-catechin, (−)-epigallocatechin gallate, (−)-gallocatechin gallate, rutin, and theaflavin, which could greatly reduce high-fat diet-induced adverse effects, via easing food stagnation, eliminating indigestion, enhancing gastrointestinal motility, and regulating the activity of related enzymes [68,69]. Meanwhile, studies have revealed that AN could increase the risk of obesity and hyperglycemia. Gut microbes have the potential to explain these two opposite effects induced by AN. Despite few studies having explored the role of gut microbes in the carcinogenicity of polyphenols, there have been several studies linking gut microbes to cancer development and treatment [70,71,72,73]. For example, intestinal *fusobacterium nucleatum* promotes colorectal cancer development and facilitates tumor metastasis and chemoresistance to 5-fluorouracil via its immunosuppressive effects [74,75,76]. Also, gut microbiota regulates the activity, efficacy, and toxicity of chemotherapy agents, such as gemcitabine [77], cyclophosphamide [78,79], irinotecan [80,81,82], and cisplatin [83,84,85]. Further research is required to clarify whether areca nut chewing changes the composition and function of intestinal microbes or whether intestinal microbes metabolize areca nut into carcinogens.

In recent years, there has been increasing evidence that the aryl hydrocarbon receptor (AhR) plays a major role in tumorigenesis and makes the AhR an interesting pharmacological target in cancer treatment [86,87,88,89]. Polyphenols, especially flavonoids, major constituents of AN, are the largest class of natural AhR ligands that are available for humans and animals [90,91]. Mounting evidence demonstrated that flavonoids, exhibiting AhR agonist and/or antagonist activity, are widely used for the regulation of the intestinal immune system and tumor treatment [92,93,94,95,96]. Dietary flavonoids are absorbed in the intestine, and the intestinal microbiota, which is deeply involved in the metabolism of them, originated from foods [97], and in turn, acting as AhR ligands, are able to regulate intestinal microbiota composition and intestinal immunity [98]. For example, tryptophan, a reported AhR ligand, could be metabolized by the certain bacterial strain, *Lactobacillus bulgaricus* OLL11816, to AhR-activating indoles that have shown AhR-activating potential [99,100]. However, whether flavonoids are dietary, generated by the host, or through bacterial metabolism has not been exactly established and requires further investigation.

In conclusion, few studies have reported the gut microbiome profiles in AN chewers, but some studies have shown oral microbiota composition alteration might mirror oral cancer progression in AN chewers [26,27]. However, whether microbial changes are involved in areca nut-induced oral carcinogenesis is only speculative. Further research is required to discern the clinical significance of an altered oral microbiota and the mechanisms of oral cancer development in areca nut chewers. Additional studies are necessary to clarify the precise metabolic intermediates of AN by gut microbiota or the single agent responsible for AN toxicity.

## 5. Addiction in Areca Nut

### 5.1. Areca Alkaloids(Arecoline)

Although some AN chewers have been aware of the harmful effects brought by AN, they cannot abolish this habit because AN quid chewing is able to produce a sense of well-being, euphoria, warm sensation of the body, salivation, palpitation, sweating and heightened alertness, combat against hunger, and increase capacity and stamina to work by its numerous central nervous system effects [101]. It has been reported that the addictive property of AN is prominently due to its alkaloids [35]. Further, various levels of areca alkaloids could potentially contribute to variations in addictive potential in AN [102,103]. Major alkaloids found in AN are arecoline, arecaidine, guvacoline, and guvacine. Major alkaloids found in AN and their activities are showed in Table 3. Among these, arecoline has deep brain penetration to exert its numerous parasympathetic and muscarinic effects, which is responsible for the addiction and habitual use of AN. Mechanistically, arecoline acts on nicotinic acetylcholine receptor (nAChR), which partly accounts for the addiction and habitual use of AN [104,105], while the present data suggests a role also played by sympathetic activation [99]. These studies demonstrated further that the effects of arecoline reached the maximal within 4–6 min and high levels of arecoline are present in the oral cavity even 10 min after the onset of AN chewing, suggesting that active compounds released from areca nut chewing are absorbed mainly in the oral cavity, most probably through the mucous membrane [106]. 

### 5.2. Microbiota-Gut-Brain Axis: A Potential Regulator of AN Addiction

In addition, the proposal of microbiota–gut–brain axis provides a novel insight into clarifying the addictive mechanism of AN (Figure 2). Gut cytokines are known to activate the vagus nerve that constitutes the main axis transferring gut microbiota information to the brain, while the latter induces pro-rewarding effects in nucleus accumbens [109]. An expanding body of evidence supports that gut microbiota modifications and/or manipulations may also play a crucial role in the manifestation of specific behavioral responses regulated by neuroendocrine pathways. The gut microbiota and their metabolites influence neuroendocrine function through several routes, including the vagus nerve; immune activation with production of immune mediators; and production of neurotransmitters, short chain fatty acids (SCFAs), and tryptophan, to modify host behaviors relevant to stress, addiction, cognition, eating, and sexual and social behavior [110]. Xu, et al. examined the composition and diversity of intestinal microbiota in patients with substance use disorders (SUDs) and in healthy controls (HCs). The results showed that the abundances of Thauera, Paracoccus, and Prevotella are significantly higher in SUDs compared to HCs [111]. Further, gut microbiota is related to excessive alcohol consumption and induces altered striatal dopamine receptor expression in a compulsive alcohol-seeking model [112]. Mounting studies have shown that cocaine and alcoholism addiction is associated with changes in the composition of the gut microbiota [113,114,115,116,117,118].

Importantly, emerging studies suggested that the gut microbiota might influence the onset/development of addictive behaviors. The role of gut microbes in alcohol dependence has been increasingly reported. Oral administration of nonabsorbable antibiotics reduces the voluntary alcohol intake in alcohol-preferring animals [109]. Individuals presented with an increased intestinal permeability and a dysbiosis might show a more severe profile of alcohol dependence than other non-dysbiosis controls [119]. Also, gut microbiota dysbiosis during chronic alcohol exposure is closely correlated with alcohol-induced neuropsychic behaviors and BDNF/Gabra1 expression [120]. The above studies provide a new perspective for understanding underlying mechanisms in alcohol addiction. Further, gut microbiota is capable of modulating alcohol withdrawal-induced anxiety in mice [121], and altering sociability and depression by inducing β-hydroxybutyrate metabolism changes in alcohol use disorder [122]. Peterson, et al. found sex-dependent associations between addiction-related behaviors and the gut microbiota composition in outbred rats [123].

Alteration of the gut microbiota in mice also drives the behavioral response to cocaine. Animals with reduced gut bacteria, by treating with prolonged treatment with non-absorbable antibiotics, showed an enhanced sensitivity to cocaine reward and enhanced sensitivity to the locomotor-sensitizing effects of repeated cocaine administration [44]. Lee, et al. showed that the gut microbiota causally mediated reward and sensory responses related to regimen-selective morphine dependence. Depleting the gut microbiota via antibiotic treatment recapitulated neuroinflammation and sequelae, including reduced opioid analgesic potency and impaired cocaine reward following intermittent morphine treatment [124]. Chronic depletion of gut microbiota also affects other behavioral and neurochemical consequences in the rat [125]. For example, Burokas, et al. showed that manipulating microbiota, thus targeting the microbiota–gut–brain axis, had anxiolytic and antidepressant-like effects and reversed the impact of chronic stress in mice [126]. In addition, oral administration of heat-killed *lactobacilli* can alter the social behavior of healthy mice [127]. The above findings provided direct evidence of the link between gut microbiota, neuroendocrinology, and addiction or other behavioral responses, highlighting the key role of the gut microbiota in the formation and treatment of drug dependence, and may provide new treatment strategies for using novel medicine targeting gut microbiota to treat drug addiction. In addition, gut microbiota might directly influence the development or onset of addiction via modulating the oxytocin, serotonin, and dopamine levels and function [110]. While the underling mechanism is required to further investigate, in order to assess which other neuroendocrine pathways are involved in addiction and in which cases the gut microbiota plays a causal role. The relationship between microbes and areca nut addiction also remains to be explored.

## 6. Conclusions and Future Perspectives

Altogether, more microbiome research remains to be conducted in the coming years in an attempt to better characterize the role of human microbiome in carcinogenicity and addiction of areca nut. AN chewing is known to be a risk factor for several diseases, it could influence the human microbiome directly and indirectly via introducing its own microbiota, suppressing the immune system, changing the local microenvironment, or other potential mechanisms. The exact explanation of how AN chewing affects the microbiome still requires further exploration.

The property of carcinogenicity and addiction in AN brings great harm to people’s health; there is no adequate explanation for it currently. However, it is known that the carcinogenicity of areca nut is mainly polyphenols and arecoline, and the latter is significantly associated with areca nut addiction. Polyphenols, originating from various natural products, have been reported to have both carcinogenic and anticancer effects, with a two-way interaction with gut microbiota. Most polyphenols are intensively catabolized by gut microbiota to a wide variety of new chemical structures, with physiological and biochemical effects. Further, polyphenols could change the gut microbiota composition and function. Establishing an adequate mechanism of how gut microbes and polyphenols interplay might help us clarify dual, contradictory property of polyphenols. Recently, the notion of microbiota–gut–brain axis provides a novel insight into clarifying addictive mechanisms of AN. Several studies have reported a significant alteration in individuals of alcohol and drug addiction. It has long been recognized that gut cytokines can activate the vagus nerve, and the latter induces pro-rewarding effects in nucleus accumbens. In addition, mounting studies suggest that the gut microbiota might directly or indirectly influence to some extent the onset/development of addictive behaviors. Despite the findings for the interaction of AN chewing addiction and gut microbiota being scarce, reported studies provide some evidence of the link among gut microbiota, neuroendocrinology, and addiction. Further investigation is required in order to assess in which cases the gut microbiota plays a causal role in AN addiction and which other neuroendocrine pathways are involved. More studies integrating metagenomics, transcriptomics, and metabolomics with clinical results are required to gain more insight into the hugely complex network of the AN chewing-microbiome-host phenotype; in addition, finding a novel intervention target to solve the carcinogenicity and addiction of areca nut is important. 

## Figures and Tables

**Figure 1 molecules-27-08171-f001:**
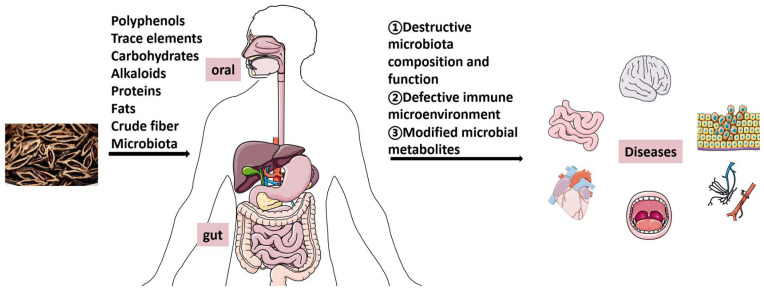
The major constituents of areca nut-induced dysbiosis of microbiome and its possible role in different diseases.

**Figure 2 molecules-27-08171-f002:**
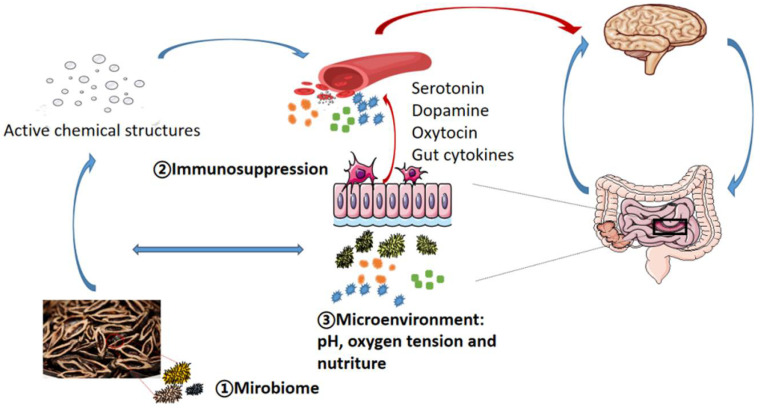
The possible mechanisms of AN-chewing-induced microbial dysbiosis and the role of gut microbiota in areca nut metabolism, as well as the direct and indirect interaction among gut microbiota, neuroendocrinology, and addiction.

**Table 1 molecules-27-08171-t001:** Prominent polyphenols in AN and their activities.

Formulas	Polyphenols	Activity	Gut Microbiota-Relevant	Reference
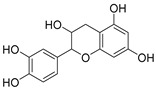	Catechin	Antimicrobial, antioxidant, anti-cancer and carcinogenic activity	Yes	[48,49]
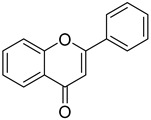	Flavonoids	Anti-inflammatory, antioxidant and anti-cancer effects, anti-bacteria and anti-virus	Yes	[9,50]
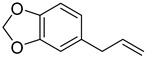	Safrole	Anti-cancer and antioxidant effects	Yes	[51]
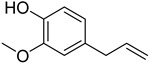	Eugenol	Anti-bacteria and antihypertensive effect	Yes	[52]

**Table 2 molecules-27-08171-t002:** Other reported major classes of polyphenols extracted from plants and their activities.

Polyphenols	Representative Plant	Activity	Gut Microbiota-Dependent	Reference
Resveratrol	*Reynoutria japonica* Houtt. and *Vitis*	Antimicrobial, antioxidant, and anti-inflammatory activity	Yes	[53]
Quercetin	*Fagopyrum esculentum* Moench.	Anti-virus, anti-bacterial, anti-cancer, and cardiovascular-protective effect	Yes	[54]
Catechin	Green tea	Anti-cancer, anti-virus, anti-fungi, anti-bacterial, and cardiovascular-protective effect	Yes	[55]
Puerarin	*Pueraria lobata*	Anti-oxidant, anti-inflammatory, antihypertensive, and neuroprotective activity	Yes	[56]
Anthocyanidin	Elderberries	Anti-oxidant, anti-mutagenic, and anti-proliferative properties	Yes	[57]
Tannic acid	Fruit	Anti-oxidant and anti-bacterial effect	Yes	[58]

**Table 3 molecules-27-08171-t003:** Major alkaloids found in AN and their activities.

Formulas	Alkaloids	Activity	Reference
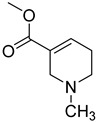	Arecoline	Effects on nervous, cardiovascular, endocrine and digestive system; anti-parasitic effects; carcinogenic; and genotoxic	[90]
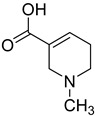	Arecaidine	Effects on nervous and endocrine system	[107]
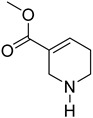	Guvacoline	Effects on nervous system, anti-inflammatory, and anti-cancer activity	[91]
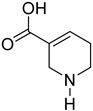	Guvacine	Effects on nervous system and anti-cancer activity	[108]

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
