# Peer review of "Potential Role of Host Microbiome in Areca Nut-Associated Carcinogenesis and Addiction"

_molecules, 2022, doi:10.3390/molecules27238171_

Round 1
Reviewer 1 Report
I have reviewed the paper entitled “Potential Role of Host Microbiome in Areca Nut3 associated Carcinogenesis and Addiction”. The paper contains very systematic information, however, needs the below modification.
Comments
1. The abstract is too short, it should be more decorated.
2. Make sure to update the introduction by citing the relevant published paper in 2020 and 2021.
3. The data provided in table 1 is not sufficient, the phytochemistry part should be revised strongly.
4. One more decorated figure should be designed.
Reviewer 2 Report
The topic is interesting, however, there are major revisions needed before the publication. As the article is about areca nut compounds, please provide table with areca nut main polyphenolic constituents - other plants could be left as "main source", but the table should deal with areca nut, not polyphenolic compounds in general. According to the data provided, the worst carcinogenic and addictive effects of areca nut are due to alkaloids present in this plant. Please provide formulas of the main alcaloids and describe their effects if known.
Extensive English editing is needed, as there are spelling errors left, e.g. L36 - Austrilia, etc.
Round 2
Reviewer 2 Report
Please reorganize the table 1 in the readable way, that all the formulas provided are readable
Please redraw the table 2 as the table is presented as major classes of polyphenolic compounds, then leaves and fruits are given in the table as them? The first column is plant, then resveratrol is given as plant?
Why do you state that it is impossible to provide formulas of main alcaloids, as they are freely available on PubChem. Please, make a figure using their formulas:
Arecoline:
https://pubchem.ncbi.nlm.nih.gov/compound/Arecoline
Arecaidine:
https://pubchem.ncbi.nlm.nih.gov/compound/10355
Guvacoline:
https://pubchem.ncbi.nlm.nih.gov/compound/160492
Guvacine:
https://pubchem.ncbi.nlm.nih.gov/compound/3532
